# The ECETOC-Targeted Risk Assessment Tool for Worker Exposure Estimation in REACH Registration Dossiers of Chemical Substances—Current Developments

**DOI:** 10.3390/ijerph17228443

**Published:** 2020-11-14

**Authors:** Jan Urbanus, Oliver Henschel, Qiang Li, Dave Marsh, Chris Money, Dook Noij, Paul van de Sandt, Joost van Rooij, Matthias Wormuth

**Affiliations:** 1Shell Health Risk Science Team, Belgian Shell N.V., B-1000 Brussels, Belgium; 2Corporate Health Management, BASF SE, 67056 Ludwigshafen am Rhein, Germany; oliver.henschel@basf.com; 3Clariant Produkte (Deutschland) GmbH, 65843 Sulzbach am Taunus, Germany; qiang.li@clariant.com; 4ExxonMobil Biomedical Sciences Inc, ExxonMobil, Leatherhead KT22 8UX, UK; dave.marsh@exxonmobil.com; 5Cynara Consulting, Brockenhurst SO42 7RX, UK; chrismoneyuk@gmail.com; 6In Personal Capacity, Formerly Dow Global Industrial Hygiene Expertise Centre, 4531 EB Terneuzen, The Netherlands; dnoij@zeelandnet.nl; 7Shell Health Risk Science Team, Shell International B.V., 2596 HR The Hague, The Netherlands; Paul.vandeSandt@shell.com; 8Caesar Consult, 6503 CB Nijmegen, The Netherlands; Joost.vanrooij@caesar-consult.nl; 9Syngenta Crop Protection AG, 4058 Basel, Switzerland; Matthias.Wormuth@syngenta.com

**Keywords:** chemicals, exposure modelling, occupational exposure, chemical regulation

## Abstract

(1) Background: The ECETOC Targeted Risk Assessment (TRA) tool is widely used for estimation of worker exposure levels in the development of dossiers for REACH registration of manufactured or imported chemical substances in Europe. A number of studies have been published since 2010 in which the exposure estimates of the tool are compared with workplace exposure measurement results and in some instances an underestimation of exposure was reported. The quality and results of these studies are being reviewed by ECETOC. (2) Methods: Original exposure measurement data from published comparison studies for which six or more data points were available for each workplace scenario and a TRA estimate had been developed to create a curated database to examine under what conditions and for which applications the tool is valid or may need adaptation. (3) Results: The published studies have been reviewed for completeness and clarity and TRA estimates have been constructed based on the available information, following a set of rules. The full review findings are expected to be available in the course of 2021. (4) Conclusions: The ECETOC TRA tool developers periodically review the validity and limitations of their tool, in line with international recommendations.

## 1. Introduction

When the European chemicals control legislation “REACH” (Registration, Evaluation, Authorization and Restriction of Chemicals) came into force in 2007 a period of just over 3 years was allowed to registrants, primarily industrial companies, to submit dossiers for a large first number of substances that were either being produced in volumes of more than 1000 tonnes per year or were classified for severe health or environmental effects [1].

This period of 3 years was relatively short considering the novel nature of REACH and required substantial work effort from companies to comply with the new regulation. In the newly formed consortia required by REACH, companies had to compile and agree on substance hazard information and identify, as well as describe, uses of the substances across often complex downstream supply chains. In addition, safe use during the life cycle of the substance had to be demonstrated if the substance was classified for any health, environmental, or physical hazard. The penalty of missing the registration deadline of 30 November 2010 was significant, namely “no data, no market” according to REACH Article 5, and the continuation of manufacture or import was forbidden if a registration had not been completed.

There were several thousand existing marketed substances that fell into the first phase. The identified health hazards of these substances for which use-specific chemical safety assessments had to be developed ranged from relatively minor, such as eye irritation, to serious such as carcinogenic or mutagenic properties. In view of the size of the challenge of identifying the many uses and estimating human and environmental exposure throughout often complex supply chains with a variety of operational conditions, a systematic generic approach was necessary.

The European Chemicals Bureau and its successor the European Chemicals Agency, ECHA, in consultation with industry and member state stakeholder partners introduced a new system of codified descriptors for environmental release categories, worker activities, and consumer product categories to enable a considerable degree of simplification and standardization [2]. In this system the worker activity coding was done with an initial list of 25 Process Categories (PROCs), extended to 28 in a later version of the guidance, which represent handling and applications of chemical substances and products commonly encountered across many industrial and professional settings. Similar simplified and standardized coding systems have been proposed in other settings, for example by Marquart et al. [3] in the development of the Advanced REACH Tool for worker exposure modelling and by the United States Environmental Protection Agency for use in substance risk assessment for specific sectors of industry [4,5].

Besides the use of PROCs for the systematic assessment of exposure and the communication of chemical substance uses up and down supply chains, the system allows for coding using combinations of PROCs such as those available at the level of industry sector groups. As an example, the European solvents industry group has developed a set of generic exposure scenarios (GES) that describe, both colloquially and in REACH terms, the nature of activities encountered in the sector, since typical use of product containing a solvent could encompass between 5 and 8 PROCs [6]. The generic approach reflects the vast majority of uses of the substance, including delivery, storage, formulation, cleaning, and maintenance.

The European Centre for Ecotoxicology and Toxicology of Chemicals (ECETOC) created the targeted risk assessment (TRA) tool to enable the estimation of worker, consumer, and environmental exposure by means of three separate tool modules [7]. The TRA for workers estimates both inhalation and dermal exposure and was originally developed accounting for the method of exposure predictions contained in the Estimation and Assessment of Substance Exposure (EASE) model previously used by European regulatory authorities for chemical safety assessment [8], and subsequently adapted to the PROC system.

Starting from the volatility or dustiness of a chemical substance, i.e., the fugacity or “tendency to become airborne”, EASE provided exposure estimates modifiable for a set of operational conditions (OC), such as the level of process containment, and risk management measures (RMM) in place such as local exhaust ventilation which had to be specified by the exposure assessor. The TRA tool took this approach one step further by consolidating some of these OC/RMM combinations under a PROC base estimate, to be then further modified for: (1) The percentage of the chemical substance in the product; (2) the duration of the exposure; (3) the presence of room and/or local exhaust ventilation; and (4) any personal protective equipment in use to limit inhalation or dermal exposure. These basic modifying factors were aligned with broad bands of fugacity, percentage of substance, and duration.

The TRA provided for each PROC and volatility or dustiness combination two distinct base estimates: the first one for “industrial use,” reflecting a good basic standard of occupational hygiene in line with the typical legal requirements of Member States of the European Union, and the second one for “professional use” where encountered exposures are often higher because of less established systems of work, exposure control and levels of supervision, education, and training. While the EASE output was given as a range intended to reflect the interquartile values of a distribution, the TRA took the upper value only in order to provide a single value, therefore nominally reflecting the 75th percentile of an underlying distribution.

At the time of the first REACH registration deadline in 2010, TRA version 2 was in use; subsequently, a revised version 3 was launched in 2011 that allowed some further refinements in the exposure estimation based on the frequently encountered needs of exposure assessors [9,10]. TRA version 3 also formed the basis for the exposure estimation algorithms in ECHA’s new platform CHESAR for the creation of Exposure Scenarios for registration dossiers.

In addition to the registrations of high-volume substances in 2010, many substance registration dossiers for chemical substances manufactured or imported in smaller quantities by the subsequent legal deadlines in 2013 and 2018 have made use of the TRA. The reason is the advantage of a simple, “lower Tier” tool which requires only a few data inputs but also provides generic exposure estimates. In contrast, higher tier tools often require more input details such as information on room size and ventilation rate. This information may be challenging to realistically specify across a large variety of workplace situations or even impossible to obtain because of the lack of access or contacts throughout the supply chain. The same applies to measurement data sets often portrayed as the “gold standard” for chemical regulatory exposure assessment purposes.

REACH, as a European Regulation, is applicable across all countries of the European Economic Area and use descriptions in registration dossiers for chemical substances are expected to reflect this. It is clearly a difficult task to provide valid exposure estimations applicable for such a wide geographical region like the EU, stretching from Galway in the West to Limassol in the East and from Lampedusa in the South to Kuopio in the North, with not only varying climatic conditions, but also differing in their national legally required worker health protection policies and their implementation.

Clearly, exposure assessors need to be aware of the validation status of the models they use. However, in the field of occupational exposure assessment there is presently no established standard for model validation procedures. Since the publication of TRA version 2 in 2009 and then TRA version 3 in 2011, several groups of European researchers have undertaken studies to explore the validity of the TRA exposure estimates [11,12,13,14,15,16,17,18,19,20,21]. Of particular note, the German regulatory agency Bundesanstalt für Arbeitsschutz und Arbeitsmedizin (BAuA) completed a large project (the Evaluation of Tier 1 Exposure Assessment Models under REACH or ETEAM) in 2016 with the aim of testing the validity of the TRA and other “Tier-1” tools used in REACH registrations [22]. The worker exposure component of the TRA has also attracted interest from countries outside the European Union, as evidenced by validation studies from Japan, South-Korea, and the United States [23,24,25,26].

These validation studies typically make use of existing sets of worker exposure measurements, based on the assumption that these are indicative of the “true” exposure distributions existing at the workplaces. However, measurement strategies underlying these data sets can have considerable impact on the distribution descriptors such as central tendency and standard deviation. In particular, it is quite common to stratify worker activities according to exposure potential if the objective of the measurements is to test compliance with an occupational exposure limit, and to focus on operations or individuals considered likely to have elevated exposure. The resulting measurement data derived in those circumstances may exceed the true data distribution. Unfortunately, detailed information on the underlying measurement strategy for many existing and available data sets is often not available or the strategy is not well described thus introducing additional uncertainty in the interpretation of results. Using such data sets for comparison with the outcome of any exposure modelling tool like ECETOC TRA as the basis for testing the performance of the model therefore requires careful interpretation.

Despite the different initiatives and research projects, the overall TRA-performance has not yet been fully evaluated across all combinations of PROCs and substance types. Because the base estimates for inhalation and dermal exposures are not numerically related across the PROCs, a full validation of the TRA would need to cover many different exposure situations and none of the published projects has come close to this. Therefore, published statements such as “the TRA underestimates” or, conversely, “the TRA is conservative” cannot be verified by the partial validation publications available so far but need to be qualified as to which combinations of all parameter used by the program, thus PROC, setting, substance type, volatility or dustiness, and substance application were actually studied. Recently, Spinazzè et al. mapped published validation studies against PROCs, but without indicating the distinction between vapor bands of volatile liquids or dustiness bands of handled solids [27]. Regardless, the overview shows several remaining gaps. As for all exposure models used by exposure assessors for REACH dossier creation, further validation and refinement of the TRA is needed and is best done working with high quality, representative data sets of measured exposures [28]. This paper provides an interim update of work ongoing in ECETOC.

## 2. Materials and Methods

An ECETOC task force of experienced exposure assessors (authors of this paper) was established to systematically review the available publications on the performance of TRA and determine the validity of the exposure estimation by the tool based on the available evidence. This task has been undertaken in consultation with a group of external advisors and in parallel with an ongoing effort by ECHA to address the full suite of occupational exposure-modelling tools recommended for REACH chemical safety assessment which was initiated in 2018 [29]. Currently, the task force is developing a curated inventory of measurement data used by the researchers studying the TRA validity. While all those available data sets were inserted into the database, only those data were used for further analysis if they fulfill the following requirements: (1) A minimum of six measurements in order to calculate the 75th percentile and the associated confidence interval for comparison with the exposure estimate generated by the tool; (2) clear description of the chemical substance(s) and work activity being monitored so that a PROC can be assigned; (3) sufficient contextual information to define the tool inputs (modifying factors) to generate a TRA estimate. While some of the published validation studies relied on measurement data sets of only 1–3 samples for some of their conclusions, the ECETOC TRA task force focused, in line with the recommendations of, in particular, EN 689 [30], on measurement data sets of a minimum of six samples. This is considered to provide a more robust characterization of the distribution of the worker exposure and a reasonable basis for comparison with the modelled exposure estimate. In some cases, the publishing researchers have been contacted by members of the ECETOC Task Force to augment published information, such as TRA calculation assumptions, to remove ambiguities, and ensure transparency in TRA P75 calculations. Where possible, full exposure survey reports have been retrieved.

The TRA review by the task force will also benefit from the findings of several research projects commissioned in recent years under the Long-Range Research Initiative of the European chemical industry council CEFIC [31,32], including an update of the RMM library published previously as ECEL [33] and now configured to align with the TRA to support in particular the PROC-specific efficacy figures for local exhaust ventilation (LEV).

## 3. Results

The curated database is currently being finalized with full transparency as to the original measurement data, modeling approach followed by the researchers and any modifications deemed necessary by the ECETOC TF. Although it is derived from the same data sources as used in the published comparison studies, the application of some additional inclusion criteria led to the deselection of part of the data due to ambiguities in original data description that preempted the choice of the appropriate PROC or the derivation of a reliable TRA estimate, or simply due to small data sets. The following three examples illustrate the approach taken. These examples are considered of sufficient quality and clarity in description of the performed measurements and have been therefore included in the database for further analysis, either in their original form or with adaptations.

The first example is from the study by Lee at al. [24]. It concerns an industrial printing operation which utilizes isopropanol, a volatile liquid in the TRA medium volatility band (0.5–10 kPa). Additional information was obtained via e-mail from the corresponding researcher regarding detailed workplace description, ventilation conditions, chemical use rate and other operational conditions. Thirteen worker exposure measurement results, with duration between 320 and 572 min, were available. The geometric mean of 59.3 mg/m^3^ (24.1 ppm) and geometric standard deviation of 1.44 of the measurement results were used to calculate a 75th percentile of 30.9 ppm. The available contextual information was used by the ECETOC TF to derive a TRA exposure estimate of 35 ppm for a PROC 10 situation (low energy spreading) with the following conditions: substance concentration band: >25%; exposure duration: >4 h; indoor location; good general ventilation; no local exhaust ventilation. It must be noted that Lee et al. followed the approach previously adopted in the E-team project [14] in which every single measurement result was modelled against a TRA estimate. The data from the printing operation for PROC 10 had been merged with data from other operations assigned to the same PROC, hence a direct comparison with the results as presented in the original publication is not possible.

The second example comes from the work by Jankowska and colleagues [17] and concerns the use of the anesthetic Sevoflurane, a high volatility liquid, in an operating theater. Forty-two exposure measurement results were reported from which the ECETOC TF calculated a 75th percentile of 0.33 ppm. The measurement duration was indicated as 6 h but the actual exposure duration was reported as approximately 2 h. The ECETOC TF derived a TRA estimate of 0.84 ppm, based on: a professional setting; a PROC 2 situation (closed process with occasional controlled exposure); TRA concentration band 1–5%; good general ventilation and local exhaust ventilation; and exposure duration of 1–4 h. Jankowska et al. in their research interpreted their calculated TRA estimate as a 90th percentile and derived a TRA estimate of 0.86 ppm (7 mg/m^3^) versus the figure of 0.97 ppm (7.96 mg/m^3^) from their measurements and concluded that the TRA underestimated exposure in this specific scenario. The ECETOC TF derived essentially the same TRA estimate but interpreted the measurement information differently on the following aspect: the tool output represents a 75th percentile, hence the exposure measurement data set needs to be processed slightly differently.

The third example concerns a data set reported by Lee and colleagues for seven scenarios with exposures to allyl alcohol or acetone in Korea [25]. According to the information presented, workers wore respiratory protective equipment. In their calculations of the TRA estimates for comparison with the exposure measurement results the Korean researchers included the exposure reducing effect of the respirators. In the view of the ECETOC TF this is incorrect, as it is highly unlikely that exposure has been measured inside respirators: not only is this impractical and would go against commonly adopted industrial hygiene standards, it would possibly also invalidate the respiratory protection by breaking the seal between respirator and the workers’ face. In an e-mail correspondence the authors/researchers declared that they have not visited any of the workplaces in their study and observed any actual monitoring. Second, under the Korean Occupational Safety and Health Act occupational exposure measurements are taken outside respirators. In the curated database these data sets have been included without accounting for the respirator effect.

## 4. Discussion

Overall, analysis of exposure model performance based on comparison with measurement data sets is a work-intense exercise which requires careful review of data and information and a clear set of rules regarding acceptance of presented evidence. Sufficient detail on the conditions and circumstances of occupational use of substances is needed to select the model inputs unequivocally, both for routine use of the tool in exposure estimation as well as under the current scope of a validation review. Whereas in the case of exposure estimation the absence of some information elements can be countered by making conservative assumptions, for a validation review it is essential to take well-described situations with a sufficient number of exposure measurements to characterize the exposure distribution and derive a robust percentile figure for comparison with the modelling tool output. As the ECETOC TRA is a so-called Tier-1 tool the inputs are deliberately fewer in number. Nevertheless, ambiguities in relation to actual worker activity and location with respect to sources of exposure, status of general and local ventilation, composition of products containing the chemical of interest and other operational conditions should be avoided, as these may significantly impact the conclusions drawn.

The use of existing exposure measurement study reports to validate an exposure model, while attractive from the perspective of the required effort to create the basis for comparison studies, brings with it almost inevitably a degree of ambiguity that needs to be overcome. In that respect it is interesting to note that for the use of a model to estimate chemical exposure, ambiguity is addressed with conservative model inputs potentially leading to overestimation of actual exposure by the modelling approach as the objective is to define the operational conditions and risk management measures required for safe use. For an assessment of the validity of a model, in contrast, one should err on the other side, i.e., avoid drawing a conclusion of model validity if remaining uncertainties may potentially lead to underestimation by the modelling approach.

One example of such an ambiguity in the available information is where in a workplace investigation report reference is made to the presence of a local exhaust ventilation (LEV) system, but the investigators conclude in their report that the LEV is not delivering to its protective potential, because it is either incorrectly designed, positioned, used or maintained, and hence a recommendation is provided to the workplace manager to improve the LEV. Such a situation was encountered in a Health Hazard Evaluation report by the US National Institute of Occupational Safety and Health [34], part of the project reported by van Tongeren et al. [14]. The TRA exposure estimate elaborated by the ECETOC TF for comparison with the measurement data of this investigation does therefore not contain the LEV effect.

There is no established standard for validation of occupational exposure models on the basis of exposure measurement results, but some form of good practice standards can be envisaged. The present work by ECETOC aims to contribute to the development of these standards by being fully transparent on data treatment and by sharing its adopted rules for data acceptance.

The two possible outcomes of the review by the ECETOC TF are: (1) The analysis validates the existing tool (and no updates are warranted) or; (2) the analysis identifies areas to improve the model, such as changes to some base estimates and/or exposure modifiers in the TRA. The outcomes will be published, either in the form of a communication to registered users or as a new version of the tool, in the course of 2021.

Regardless, prospective research projects are clearly needed in which workplace exposure measurement surveys are undertaken utilizing a questionnaire approach that captures unambiguously all pertinent information necessary for elaborating exposure model estimates. In addition, such research projects need to ensure that a sufficient number of samples is taken to display representative exposure distributions and allow the generation of a statistically robust figure such as the 75th percentile in the case of the TRA. Such research should aim at covering the full application scope of the models. In that respect the coverage of the overall performance of TRA is still incomplete as indicated by Spinazzè and colleagues [27] and further studies are welcomed by ECETOC.

## 5. Conclusions

Availability of broadly accepted worker exposure estimation tools is essential to perform broad exposure assessments for screening purposes under chemicals control legislation such as REACH that covers the full life cycle of substances and where direct measurement of exposures in all scenarios is seldom possible. Tier-1 screening models such as ECETOC’s TRA, which require limited inputs, play an important role in managing the workload associated with substance dossier development and helping to set priorities for substance use scenarios requiring more detailed exposure assessment. Tool developers have a responsibility to provide sufficient information on the tool’s basis, its applicability domain and validity, whereas researchers are encouraged to publish and fully document their external tool validation projects. In the absence of a formal standard for validation of occupational exposure models, it is common practice to utilize measured data from well-characterized work environments. Because of the huge variety of individual workplaces, but also due to variations of exposures between individual work environments and even over time the validation of any tool is best served using a body of measurement survey results meeting standardized criteria. ECETOC is presently finalizing such a database which will contribute to the understanding of the accuracy of the TRA outputs and the correct use of the TRA tool.

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
