# Peer review of "The ECETOC-Targeted Risk Assessment Tool for Worker Exposure Estimation in REACH Registration Dossiers of Chemical Substances—Current Developments"

_ijerph, 2020, doi:10.3390/ijerph17228443_

Round 1
Reviewer 1 Report
The estimation of exposure levels of workers is not only important for the development of dossiers for REACH registration. Therefore, a comparison between such tools like ECETOC TRA and real world data are important. A sufficient number of measurements at real workplaces is required to obtain a robust estimate of the 75th percentile.
The authors used only the datasets containing at least 6 measurements. Although the 75th percentile can be determined from 6 measured values, it can be very inaccurate. The coefficient of variation of the distribution plays an important role. According to the coefficient of variation one could, for example, calculate the ratio of the width of the corresponding confidence interval and the 75th percentile and restrict it upwards. A high coefficient of variation would therefore require more measured values.
The bibliographical information is incomplete for several references. If these references are available online, the URLs should be completed.
Author Response
Thank you for your valuable comments. Please see below our responses.
"The estimation of exposure levels of workers is not only important for the development of dossiers for REACH registration. Therefore, a comparison between such tools like ECETOC TRA and real world data are important. A sufficient number of measurements at real workplaces is required to obtain a robust estimate of the 75th percentile. The authors used only the datasets containing at least 6 measurements. Although the 75th percentile can be determined from 6 measured values, it can be very inaccurate. The coefficient of variation of the distribution plays an important role. According to the coefficient of variation one could, for example, calculate the ratio of the width of the corresponding confidence interval and the 75th percentile and restrict it upwards. A high coefficient of variation would therefore require more measured values."
[Authors' response]: this is a valuable comment which we are happy to accept. We will calculate confidence intervals for our 75th percentiles, using the geometric standard deviation instead of the coefficient of variation as you suggested. We will insert an additional phrase in the manuscript to mention this.
"The bibliographical information is incomplete for several references. If these references are available online, the URLs should be completed."
[Authors' response] Indeed, several of the references need correcting. Where available, URLs will be added.
Reviewer 2 Report
The paper is well written but it needs some clarifications:
- Introduction section, paragraph that starts with "The EuThe European Chemicals Bureau and its successor...", when the authors say: "In this system the worker activity coding was done with an initial list of some 25 Process Categories...", they have to specify this "some", that is, what are the Process Categories that represent chemical substance handling and applications commonly encountered across many industrial and professional settings.
- The same paragraph when the authors say: "...by Marquart et al. [3]in the...". There has to be a white space between [3] and in.
- Introduction section, paragraph that starts with "The European Centre for Ecotoxicology and Toxicology of Chemicals...", when the authors say: "...contained in the EASE model previously in use by European regulatory..., this "in use", I think that it has to be "used".
- Introduction section, paragraph that starts with "The TRA provided for each PROC and volatility or dustiness combination two separate base estimates:". I don't understand this sentence. Is it correct?
- Introduction section, paragraph that starts with "These validation studies typically make use of existing sets..." when the authors say: "...hese data sets can have considerable impact on the distribution parameters such as...", I think it is not "the distribution parameters" but "the distribution of the parameters."
- Introduction section, paragraph that starts with "Despite the different initiatives and research projects,...", when the authors say: "Because the PROC base estimates for inhalation and dermal exposures are independent of each other, ...", what PROC base estimates? They have to specify.
- The same paragraph, when the authors say: "...TRA is conservative’ cannot be verified by the partial validation publications available so far", these publications, I think it would be parameters?
- Materials and Methods section, in the first paragraph when the authors say: "...on the performance of TRA and determine he validity of the exposure estimates...",
I think it would be: "...on the performance of TRA and determine he validity of the exposure estimation..."
- In the same paragraph they say that there is the following requeriment: "a minimum of 6 measurements in order to calculate the 75 th percentile;" Why 75 percentile and not another one?
Author Response
The paper is well written but it needs some clarifications: [Authors: Thank you]
- Introduction section, paragraph that starts with "The European Chemicals Bureau and its successor...", when the authors say: "In this system the worker activity coding was done with an initial list of some 25 Process Categories...", they have to specify this "some", that is, what are the Process Categories that represent chemical substance handling and applications commonly encountered across many industrial and professional settings. [Authors response: OK, minor text correction. ECETOC TRA covers 25 PROCs; later some extra PROCs were added to the ECHA guidance but these do not have not been added to the TRA.]
- The same paragraph when the authors say: "...by Marquart et al. [3]in the...". There has to be a white space between [3] and in. [Authors: OK.]
- Introduction section, paragraph that starts with "The European Centre for Ecotoxicology and Toxicology of Chemicals...", when the authors say: "...contained in the EASE model previously in use by European regulatory..., this "in use", I think that it has to be "used". [Authors: Will re-phrase.]
- Introduction section, paragraph that starts with "The TRA provided for each PROC and volatility or dustiness combination two separate base estimates:". I don't understand this sentence. Is it correct? [Authors' response: We will add ‘for the industrial and professional domains as they have been defined in the REACH Use Descriptor guidance’.]
- Introduction section, paragraph that starts with "These validation studies typically make use of existing sets..." when the authors say: "...hese data sets can have considerable impact on the distribution parameters such as...", I think it is not "the distribution parameters" but "the distribution of the parameters." [Authors' response: Parameters may be replaced by the word 'Descriptors'].
- Introduction section, paragraph that starts with "Despite the different initiatives and research projects,...", when the authors say: "Because the PROC base estimates for inhalation and dermal exposures are independent of each other, ...", what PROC base estimates? They have to specify. [Authors' response: OK, will add clarifying text.]
- The same paragraph, when the authors say: "...TRA is conservative’ cannot be verified by the partial validation publications available so far", these publications, I think it would be parameters? [Authors' response: No, this is about the publications. Will review text for clarity.]
- Materials and Methods section, in the first paragraph when the authors say: "...on the performance of TRA and determine he validity of the exposure estimates...",
I think it would be: "...on the performance of TRA and determine he validity of the exposure estimation..." [Authors' response: will clarify text.]
In the same paragraph they say that there is the following requeriment: "a minimum of 6 measurements in order to calculate the 75 th percentile;" Why 75 percentile and not another one? [Authors' response: The TRA was based on a previous tool which provided bands of concentrations characterised as inter-quartile intervals. The TRA used the upper values when developed in 2004, hence 75th percentiles. This is indicated in the text. Indeed it is now common to promote the 95th or 90th percentile in risk assessment, although this is often not well explained.]
Reviewer 3 Report
It is not clearly mentioned but the authors are supposed to be some of the members of the ECETOC taskforce, and if so, it should be clearly
- It is not clearly mentioned but the authors are supposed to be the members of the ECETOC taskforce, and if so, it should be clearly mentioned.
- This paper is an interim report on the work activities for the ECETOC TRA tool, and the outline of the database being created for the validation of TRA is not concretely described. So, the present form like the original research article may not be appropriate. (i.e. introduction, material and methods, results, discussion, and conclusion)
- As for abbreviations, they should be indicated when the word appears for the first time in non-abbreviated full-spelled form. Please confirm as for LEV and RMM.
- The text of this paper is somewhat difficult to understand for readers who are unfamiliar with European chemical regulations and risk assessment tools in the workplace. It may be easier to understand if you do not use abbreviated words a lot.
- Reference [30]: Insufficient description of authority / citation source.
- Please give information on “e.g. EN 689” by showing appropriate reference.
Author Response
- It is not clearly mentioned but the authors are supposed to be the members of the ECETOC taskforce, and if so, it should be clearly mentioned. [Authors' response: We will mention this.]
- This paper is an interim report on the work activities for the ECETOC TRA tool, and the outline of the database being created for the validation of TRA is not concretely described. So, the present form like the original research article may not be appropriate. (i.e. introduction, material and methods, results, discussion, and conclusion). [Authors' response: Indeed, we had originally presented this as a 'Short Communication', but the Journal editors have requested us to extend it and align with the recommended structure.]
- As for abbreviations, they should be indicated when the word appears for the first time in non-abbreviated full-spelled form. Please confirm as for LEV and RMM. [Authors' response: Of course, we will make sure this convention is applied.]
- The text of this paper is somewhat difficult to understand for readers who are unfamiliar with European chemical regulations and risk assessment tools in the workplace. It may be easier to understand if you do not use abbreviated words a lot. [Authors' response: OK.]
- Reference [30]: Insufficient description of authority / citation source. [Authors' response: Will check.]
- Please give information on “e.g. EN 689” by showing appropriate reference. [Authors' response: will remove e.g. and refer to EN 689 properly.]
This manuscript is a resubmission of an earlier submission. The following is a list of the peer review reports and author responses from that submission.